# Uni²Det: Unified and Universal Framework for Prompt-Guided Multi-dataset 3D Detection

**Yubin Wang**[1*], **Zhikang Zou**[2*], **Xiaoqing Ye**[2], **Xiao Tan**[2], **Errui Ding**[2], **Cairong Zhao**[1†]

[1]School of Computer Science and Technology, Tongji University, [2]Baidu Inc.

## Abstract

We present Uni²Det, a brand new framework for unified and universal multi-dataset training on 3D detection, enabling robust performance across diverse domains and generalization to unseen domains. Due to substantial disparities in data distribution and variations in taxonomy across diverse domains, training such a detector by simply merging datasets poses a significant challenge. Motivated by this observation, we introduce multi-stage prompting modules for multi-dataset 3D detection, which leverages prompts based on the characteristics of corresponding datasets to mitigate existing differences. This elegant design facilitates seamless plug-and-play integration within various advanced 3D detection frameworks in a unified manner, while also allowing straightforward adaptation for universal applicability across datasets. Experiments are conducted across multiple dataset consolidation scenarios involving KITTI, Waymo, and nuScenes, demonstrating that our Uni²Det outperforms existing methods by a large margin in multi-dataset training. Notably, results on zero-shot cross-dataset transfer validate the generalization capability of our proposed method. Our code is available at `https://github.com/ThomasWangY/Uni2Det`.

## 1 Introduction

With the ability to capture precise geometric information of entire scenes, LiDAR has become an essential sensor for most autonomous vehicles. Due to the rapid development of large-scale annotated 3D LiDAR datasets such as Waymo (Sun et al., 2020), nuScenes (Caesar et al., 2020), and KITTI (Geiger et al., 2012), LiDAR-based models play a significant role in various critical perception tasks for autonomous vehicles, particularly in 3D object detection. Recent studies (Lang et al., 2019; Deng et al., 2021; Shi et al., 2020a; 2023; Chen et al., 2017; Wei et al., 2022; Yin et al., 2021; Wang et al., 2022) have made significant advancements in 3D detection using large-scale benchmarks and have demonstrated superior performance by leveraging precise 3D geometric information extracted from point clouds. However, despite these breakthroughs, current LiDAR-based models typically adhere to a paradigm of training and testing within a single dataset, which limits the source data to a narrow domain, as shown in Figure 1(a). Deploying dataset-specific models directly onto other datasets equipped with different LiDAR systems often leads to significant performance degradation due to substantial domain shifts (Yang et al., 2021; 2022). Consequently, the single-dataset paradigm fails to produce a robust and generalizable perception model, leading to poor performance on different datasets and further impairing the generalization ability.

The availability of vast training data in 2D vision (Goyal et al., 2021; Kirillov et al., 2023; Wang et al., 2023) has facilitated research into joint training of unified detectors for 2D perception tasks. However, 3D vision has not fully benefited from these advancements due to significant cross-dataset discrepancies. Thus, further exploration of multi-dataset training strategies in 3D perception tasks, particularly 3D object detection, is urgently needed. A direct approach to designing a unified 3D object detection framework for achieving multi-dataset training (MDT) involves merging multiple datasets and retraining the baseline detector on the merged dataset. However, significant domain gaps exist between 3D datasets, and directly combining multiple data sources can result in negative transfer. Some efforts (Zhang et al., 2023) focused on 3D multi-dataset object detection have offered

---

*Equal Contribution. This work was done during Yubin Wang's internship at Baidu Inc.

†Corresponding Author. Email: zhaocairong@tongji.edu.cn

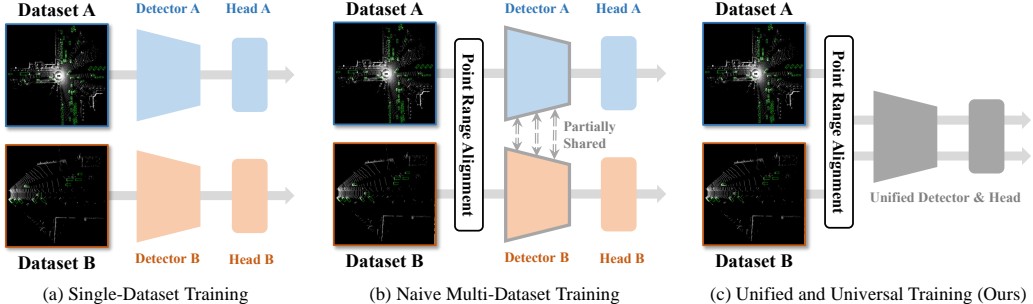

Figure 1: Illustration of different training paradigms. Single-dataset training leverages separate detectors and heads for different datasets. Naive multi-dataset training conducts point range alignment and partially shares the parameters within detectors, but still with dataset-specific heads. We propose unified and universal training, where detectors and heads for different datasets are fully shared.

solutions for building a unified training paradigm for point cloud data from different domains. As shown in Figure 1(b), the overall framework is designed in a dataset-specific manner, sharing certain backbone parameters while employing separate normalization and head layers for different datasets. Despite alleviating the unavoidable data-level differences to some extent, this independent paradigm suffers from two challenges: (i) this paradigm inhibits the full mutual utilization of each dataset's unique features, thereby constraining the further enhancement of the model's capabilities; (ii) the capacity for generalization to unseen domains is constrained due to the customization of certain network parameters specific to the trained dataset. There is a scarcity of research on both refining the unified multi-dataset training paradigm and improving its generalization to other 3D datasets in more real-world scenarios. Our main goals include effectively unifying the processing of diverse larger-scale point cloud data and ensuring robust generalization of the trained model to unseen domains. Compared to other works, we focus on developing universal techniques for managing out-of-domain datasets. Accordingly, we define our universality as the ability of a model to be jointly trained on a set of specific datasets and to perform zero-shot detection on new datasets using corresponding dataset attributes as prompts, without the need for re-training.

To achieve these goals, we propose **Uni**fied and **Uni**versal framework for 3D **Det**ection (Uni$^2$Det), which integrates multi-stage prompting modules applicable to any LiDAR dataset and various 3D object detection baselines used in autonomous driving. As shown in Figure 1(c), our approach enhances performance by learning parameters shared across datasets, referred to as unified and universal training. It utilizes diverse prompts, such as intrinsic dataset attributes that are easily obtained from target datasets but cannot be automatically learned. Due to inherent discrepancies in large-scale 3D datasets, we perform point distribution correction during voxelization to learn unified point and voxel representations across datasets, centered on mean-shifted batch normalization. Furthermore, handling data with varying statistical distributions within the backbone remains a challenging problem. To mitigate variations in data distribution, particularly from the perspective of point range, we introduce BEV-based range masking that acts on BEV features. This approach provides prior signals for the 2D convolutional network, enabling it to effectively handle point clouds from different datasets in a unified manner. Additionally, we observe that the same category exhibits statistical differences across datasets, which hinders the effectiveness of a universal detection head to some extent. To this end, we learn object-conditional residuals as prompts acting on each RoI feature, integrating features from pre-trained heads with new knowledge about the target domain. Our method does not simply combine disparate modules but should be regarded as an integrated system. It allows different components of the 3D detection framework to leverage dataset-specific prior prompts, compensating for inter-dataset variations without requiring multiple branches. Benefiting from it, models can fully utilize diverse datasets for joint training, thereby improving in-domain detection performance. At the same time, the prior characteristics of unseen datasets can also be leveraged within a unified network as encoded prompts, enabling better out-of-domain generalization. Furthermore, this framework facilitates seamless plug-and-play integration within various advanced 3D detection frameworks while allowing direct adaptation for universal applicability across datasets.

Our main contributions consist of three parts:

- We introduce a novel training paradigm for 3D object detection which focuses on unified and universal multi-dataset training, aiming at enhancing the performance in MDT settings.

- We present Uni$^2$Det, a novel framework on 3D detection with multi-stage prompting modules for prompting various components in a detector including voxelization, backbone and head, enabling robust performance across diverse domains and generalization to unseen domains.

- Experiments conducted across multiple dataset consolidation scenarios involving KITTI, Waymo, and nuScenes demonstrate that Uni$^2$Det significantly outperforms existing methods in multi-dataset training, especially on the generalization capability of the model.

## 2 RELATED WORKS

### 2.1 LIDAR-BASED 3D OBJECT DETECTION

LiDAR-based 3D object detection aims to produce a collection of 3D bounding boxes along with their associated object categories using a LiDAR point cloud. Current LiDAR-based 3D object detection research (Lang et al., 2019; Shi et al., 2020a; 2019; 2023; Yan et al., 2018; Shi et al., 2020b) can be broadly categorized into point-based methods, voxel-based methods, and hybrid point-voxel-based methods. Point-based methods, such as Point-RCNN (Shi et al., 2019) and 3DSSD (Yang et al., 2020) generate feature maps directly from raw point clouds, thereby leveraging more accurate geometry information compared to previous methods. Unlike point-based methods, Voxel-based methods like VoxelNet (Zhou & Tuzel, 2018) and SECOND (Yan et al., 2018) initially voxelize the input point cloud, transforming irregular LiDAR points into ordered voxels, and then extract features using 3D convolutions. PointPillars (Lang et al., 2019) encodes the input point cloud into pillars and employs 2D convolutions for feature extraction. Voxel-RCNN (Deng et al., 2021) analyzes the advantages of voxel features and explores a balanced trade-off between detection accuracy and inference speed. Additionally, some studies attempt to merge the advantages of point-based and voxel-based representations. PV-RCNN (Shi et al., 2020a) and PV-RCNN++ (Shi et al., 2023) leverage both multi-scale 3D voxel CNN features and PointNet-based features, consolidating them into a concise set of keypoints using a newly proposed voxel set abstraction layer. Nevertheless, all the aforementioned detectors are trained and evaluated using separate 3D datasets, leading to significant degradation in detection accuracy when applied to other different datasets.

### 2.2 MULTI-DATASET TRAINING

In recent years, training on multiple diverse datasets has emerged as an effective strategy for enhancing model robustness. Multi-dataset training has been previously investigated in the image domain, particularly in tasks such as object detection (Zhou et al., 2022; Wang et al., 2019) and image segmentation (Lambert et al., 2020). For perception tasks (Dai et al., 2021; Gong et al., 2021; Zhao et al., 2020), dataset unification involves consolidating various semantic concepts. Early studies (Lambert et al., 2020; Zhao et al., 2020; Xu et al., 2020) have focused on merging taxonomy information and training models on a unified label space. MSeg (Lambert et al., 2020) manually unified the taxonomies of different semantic segmentation datasets and resolved inconsistent annotations between them. Universal-RCNN (Xu et al., 2020) trains a partitioned detector on multiple large datasets and modeled class relations using an inter-dataset attention module. To reduce the annotation cost associated with unifying the label space, recent studies (Zhou et al., 2022; Wang et al., 2019) have explored the use of dataset-specific supervision. Although joint training of a unified detector has been studied in 2D perception tasks, further exploration in 3D perception tasks, such as 3D object detection, remains urgently needed. Uni3D (Zhang et al., 2023) attempts to design a framework in a dataset-specific manner, sharing certain backbone parameters while employing separate normalization and head layers for different datasets. PPT (Wu et al., 2024) proposes a novel framework for multi-dataset synergistic learning in 3D representation learning that supports multiple pre-training paradigms with prompt-driven normalization. However, its learning of dataset-specific prompts relies on an automatic approach that does not incorporate intrinsic characteristics of the dataset, which are not sufficiently universal since it requires new parameters for each new dataset. OneDet3D (Wang et al., 2024) seeks to achieve universality by integrating indoor and outdoor datasets into a unified framework, employing an open-vocabulary strategy for object detection. Despite its merits, OneDet3D's experiments on outdoor datasets are limited exclusively to the car category rather

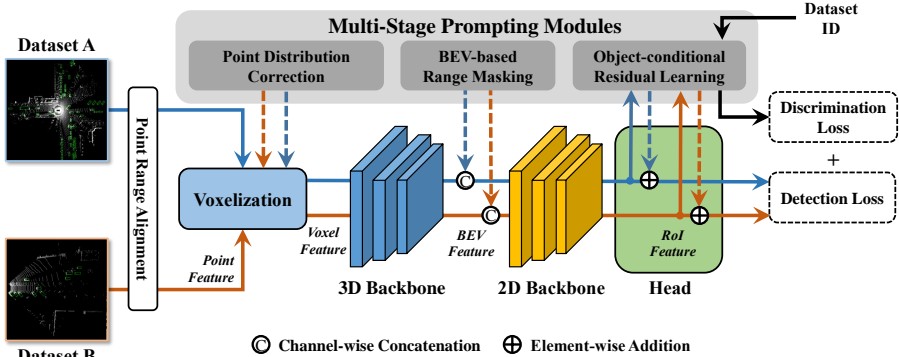

Figure 2: Illustration of the overall framework of Uni$^2$Det. The multi-stage prompting modules are employed as the core component to make the detection more unified and universal.

than utilizing a broader range of categories for both training and inference. To address these issues, we propose Uni$^2$Det for 3D detection, which integrates multi-stage prompting modules applicable to any LiDAR dataset and various 3D detection baselines, enabling robust performance across domains and generalization to unseen domains. Our method leverages the dataset's inherent properties, such as point cloud distribution and range, which are easily obtainable in other datasets and is inherently more generalizable than other methods.

## 3 METHOD

The overall framework is shown in Figure 2. We first describe our problem setting and the multi-dataset evaluation method in Sec. 3.1. Next, we introduce our multi-stage prompt learning modules for multi-dataset 3D detection, from various components in detectors including **Voxelization** in Sec. 3.2, **Backbone** in Sec. 3.3 and **Head** in Sec. 3.4.

### 3.1 PRELIMINARY

In the realm of 3D object detection, the task involves analyzing an input frame of LiDAR points to predict associated labels, including categories and orientated bounding boxes. Training an object detection model $\mathcal{F}$ with its parameter $\Theta$ on a single dataset typically involves a straightforward approach: minimizing the 3D detection loss $\ell$ over a set of point clouds $\mathbf{x}$ and its corresponding ground truth $y$ from the dataset $\mathcal{D}$:

$$\min_{\Theta} \mathbb{E}_{(\mathbf{x},y) \in D} \left[ \ell(\mathcal{F}(\mathbf{x}; \Theta), y) \right]. \tag{1}$$

Suppose that a dataset is characterized by a joint probability distribution $P_{XY}$ over the input point cloud and label space $\mathcal{X} \times \mathcal{Y}$. In the scope of multi-dataset training (MDT), we possess $N$ datasets $\{\mathcal{D}_i\}_{i=1}^N$ originating from diverse domains. Each $\mathcal{D}_i$ is linked to a distinct data distribution $P_{XY}^i$. The goal of MDT is to utilize multiple labeled datasets for training a unified model $\mathcal{F} : \mathcal{X} \to \mathcal{Y}$, aiming for increased generalizability and minimized prediction errors across various domains. One straightforward strategy entails merging all datasets into a substantially larger one, denoted as $\mathcal{D}_{merge} = \mathcal{D}_1 \cup \mathcal{D}_2 \cup \cdots \cup \mathcal{D}_N$. While datasets may feature distinct label spaces, our training and evaluation are limited to categories relevant to autonomous driving scenarios: vehicle, pedestrian, and cyclist. Consequently, the label space $\mathcal{Y}$ can be shared across various domains. This approach optimizes the same loss function over the expanded dataset $\mathcal{D}_{merge}$:

$$\min_{\Theta} \mathbb{E}_{(\mathbf{x},y) \in \mathcal{D}_{merge}} \left[ \ell(\mathcal{F}(\mathbf{x}; \Theta), y) \right] \tag{2}$$

In the following sections, we present the design of our Uni$^2$Det and show how to train a 3D perception model that performs well on seen datasets and generalizes to unseen datasets.

## 3.2 PROMPT FOR VOXELIZATION: POINT DISTRIBUTION CORRECTION

The discrepancy in point distribution among different datasets poses a significant challenge in multi-dataset training. We use KITTI as an example. Since point features are typically related to coordinates in the ego-coordinate system, the feature channel corresponding to the x-axis consistently has values greater than 0 in KITTI. As a result, differences between KITTI and other datasets cause variations in the point cloud feature distribution, which in turn influences the training process across multiple datasets. To address data-level discrepancies in large-scale annotated 3D LiDAR datasets, we aim to develop simple modules during voxelization. These modules will enable existing 3D detectors to learn universal point and voxel representations across diverse datasets, as shown in Figure 3(a).

**Point representation learning**    Instead of relying on coordinates as point features, certain studies have explored effective methods of fusing information from various viewpoints. This is achieved through a learnable network incorporating a linear layer and batch normalization. However, in this approach, the batch normalization process does not account for MDT training scheme, where points within the batch come from frames in various datasets having large statistic differences. To address this, we introduce a new normalization approach termed "Mean-shifted batch normalization" to perform instance-level feature correction. Compatible with any 3D detectors, this method can alleviate statistical differences in features extracted by standard 2D or 3D backbones.

**Mean-shifted batch normalization**    After the linear layer, we obtain a batch of point features represented as $P = \{p_1^1, p_2^1, ..., p_j^i, ..., p_{N_M}^M\}$ from $M$ frames. Here, $p_j^i$ denotes the $j$-th point feature, and $N_i$ represents the total number of points in the $i$-th frame. Conventional BN carries out normalization across all frames (instances) under the assumption that all data follows the same distribution. However, two frames may exhibit disparate point ranges due to the different sensors in use and even if they come from the same sensor, the distribution of points may still be highly random. To address this, under the MDT setting, we argue that instance-level statistics are also crucial and introduce mean-shifted batch normalization. Subsequently, samples from each dataset are regularized using the basic mean $\mu$ with an adjustment from the current instance-specific mean $\mu^i$, as follows:

$$\hat{p}_j^i = \frac{p_j^i - \alpha\mu^i - (1-\alpha)\mu}{\sqrt{\sigma + \epsilon}}, \qquad (3)$$

where $\mu$ and $\sigma$ denote the channel-wise mean and variance of the feature set $P$, which are employed for conventional channel-wise feature normalization to ensure the input data conforms to zero-mean and unit-variance, and $\epsilon$ is added to ensure numerical stability. Here we maintain the sharing of variance $\sigma$. $\alpha \in [0, 1]$ is a balancing ratio for the shifted mean. When $\alpha = 0$, it is equivalent to performing the regular BN operation, while $\alpha = 1$, the normalization procedure disregards the basic mean and relies solely on instance-level statistics. The subsequent transformation step for $\hat{p}_j^i$ remains the same as in conventional batch normalization. This approach allows us to learn universal point and voxel representations across diverse datasets with instance-level statistics as regularization. Compared with PPT (Wu et al., 2024) and Uni3D (Zhang et al., 2023), our approach operates only at the voxelization stage, using low-level point cloud distribution as a dataset attribute to prompt and regularize features. The advantage of feature uniformity ensures that the training of subsequent model parts is not negatively affected by feature differences between datasets.

## 3.3 PROMPT FOR BACKBONE: BEV-BASED RANGE MASKING

In the realm of modeling point clouds in the MDT setting, learning unified features from diverse sources and domains poses a significant challenge due to variations in the point range and data distribution across different datasets. To address this challenge, we introduce BEV-based range masking acting on BEV features to effectively handle point clouds from different datasets, as shown in Figure 3(b).

Given the preconfigured point range $(x_1, y_1, x_2, y_2)$ for producing BEV features with an aligned coordinate system, where $x_1 < x_2, y_1 < y_2$, we can infer a binary mask for each dataset based on its point range. The purpose of this mask is to explicitly indicate whether the regions or grids on the BEV plane are inside the point range of the frame. Suppose $H$ and $W$ are the spatial shape of the

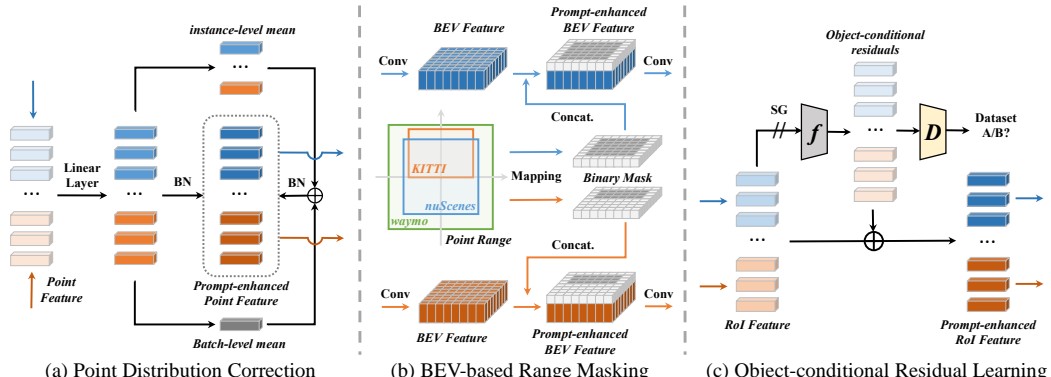

Figure 3: Illustration of multi-stage prompting modules, including three modules for prompting different components of the detector.

BEV plane, and $(x_1^i, y_1^i, x_2^i, y_2^i)$ is the point range of the $i$-th dataset. We naturally map this point range to the BEV plane based on the preconfigured point range according to the following equation:

$$x_1^{i\,'} = \lfloor \frac{(x_1^i - x_1)}{x_2 - x_1} H \rfloor, y_1^{i\,'} = \lfloor \frac{(y_1^i - y_1)}{y_2 - y_1} W \rfloor, x_2^{i\,'} = \lceil \frac{(x_2^i - x_1)}{x_2 - x_1} H \rceil, y_2^{i\,'} = \lceil \frac{(y_2^i - y_1)}{y_2 - y_1} W \rceil. \quad (4)$$

Having the mapped point range on the BEV plane $(x_1^{i\,'}, y_1^{i\,'}, x_2^{i\,'}, y_2^{i\,'})$, we can obtain mask $M^i \in \mathrm{R}^{H \times W}$ for the $i$-th dataset by:

$$M_{m,n}^i = \begin{cases} 1 & \text{if } x_1^{i\,'} \le m \le x_2^{i\,'}, \ y_1^{i\,'} \le n \le y_2^{i\,'} \\ 0 & \text{if others.} \end{cases} \quad (5)$$

Given a frame of point clouds from the $i$-th dataset, our approach concatenates BEV features with the corresponding masks $M^i$ along the feature dimension before each 2D convolutional layer. Using this prior signal, the network can effectively adapt to point clouds from various datasets, avoiding excessive focus on the area outside the relevant regions, thereby maintaining the integrity of crucial information. This integration provides a solution for the unified backbone to model features across datasets, which not only preserves dataset-specific information but also enhances the robustness and adaptability of feature modeling.

## 3.4 Prompt for head: object-conditional residual learning

The prompting modules in previous stages facilitate the framework to become more 'unified'. On the other hand, learning a general detection head is crucial in designing a 'universal' framework. Since previous works use multiple dataset-specific detection heads for prediction, such designs cannot be directly transferred to new datasets and therefore cannot be considered universal. In this section, we explore the potential of a universal detection head for predicting point clouds from diverse domains without dataset-specific branches. However, directly training a detection head on multiple datasets poses challenges. As noted in previous works, the same category exhibits statistical differences across datasets, motivating us to design prompts to mitigate the distribution gap. Consequently, we introduce object-level residual learning, inspired by (Yu et al., 2023), on RoI features, integrating them from pre-trained heads with new knowledge about the target domain, as shown in Figure 3(c). Instead of learning a set of object-agnostic task residuals, we argue that learning object-conditional residuals is more effective and transferable to unseen domains as prompts.

Given a batch of RoI features $X = \{x_i\}_{i=1}^N$ from frames of different datasets and their labels $Y = \{y_i\}_{i=1}^N$, where $y_i = j$ if feature $x_i$ is from a frame of the $j$-th dataset, we feed each RoI feature $x_i$ into a residual function $f$ to obtain object-conditional residual $r_i$. The generation process is formulated as $r_i = f(\mathrm{SG}(x_i))$, where SG indicates the stop-gradient operation to prevent hindering the regular learning of RoI features. Since the generated residuals should be relevant to the domain

or dataset to which the feature belongs, we design a discriminator $D$, implemented by an MLP, to distinguish these residuals, using the dataset ID as a prior label. The discrimination process is formulated as $\hat{y}_i = D(r_i)$. We use the cross-entropy loss to measure the discrimination loss $\mathcal{L}_{dis}$ between the predicted label set $\hat{Y} = \{\hat{y}_i\}_{i=1}^N$ and the ground truth label set $Y$, which is further added to the regular detection loss $\mathcal{L}_{det}$ as the final loss. By learning such object-conditional residuals, we can enhance the original RoI feature with prior dataset-specific characteristics by $\hat{x}_i = x_i + r_i$, and models will tend to make predictions according to a specific distribution, thus mitigating the influence of statistical differences in taxonomy across datasets.

## 4 EXPERIMENTS

### 4.1 EXPERIMENTAL SETUP

**Datasets.** Our experiments are conducted on three commonly used autonomous driving datasets: Waymo (Sun et al., 2020), nuScenes (Caesar et al., 2020), and KITTI (Geiger et al., 2012). Waymo (Sun et al., 2020) stands out as the largest dataset with over 230,000 annotated 64-beam LiDAR frames gathered from six US cities. nuScenes (Caesar et al., 2020) comprises 28,130 training samples and 6,019 validation samples collected using 32-beam LiDAR. KITTI (Geiger et al., 2012) includes 7,481 annotated LiDAR frames collected via 64-beam LiDAR. These datasets exhibit variations in data-level distributions arising from disparities in LiDAR types, geographic location of data acquisition, and variations in the definition of categorical annotations.

**Implementation details** The experiments are conducted using OpenPCDet (Team et al., 2020). Particularly, we note that differences in point cloud range significantly degrade cross-dataset detection accuracy. Therefore, we align the point cloud range of all datasets to [75.2, 75.2]m for the X and Y axes and [2, 4]m for the Z-axis. In all experimental settings, we follow Uni3D (Zhang et al., 2023) and employ the standard optimization techniques utilized by PV-RCNN (Shi et al., 2020a) and VoxelRCNN (Deng et al., 2021). For the balancing ratio $\alpha$ in our proposed mean-shifted batch normalization, we set $\alpha = 0.1$ for VoxelRCNN and $\alpha = 0.5$ for PV-RCNN. This involves using the Adam optimizer with an initial learning rate of 0.01 and implementing the OneCycle learning rate decay strategy. The network is trained across 8 NVIDIA A800 GPUs, with a total training epoch set to 30. For the experiments on Waymo-KITTI and nuScenes-KITTI consolidations, the weight decay is set to 0.01, while for Waymo-nuScenes consolidation, it is set to 0.001. We utilize only 20% of the uniformly sampled frames on Waymo dataset for model training.

**Evaluation metric.** We utilize the official tools to evaluate the performance of all baselines and our method, following (Zhang et al., 2023). For Waymo, we use Average Precision (AP) and Average Precision re-weighted by Heading (APH) for each class, based on the LEVEL 1 metric. For KITTI and nuScenes, we report Average Precision (AP) in both Bird's Eye View (BEV) and 3D over 40 recall positions, with moderate case results for KITTI. AP is evaluated with an IoU threshold of 0.7 for the car category (Vehicle on Waymo) and 0.5 for pedestrian and cyclist categories. All experimental results presented in this paper are reported on the official validation set.

### 4.2 RESULTS OF ZERO-SHOT EVALUATION ON UNSEEN DATASETS

To comprehensively assess the universal design of our framework, we perform a zero-shot evaluation on several unseen 3D datasets using different detection models. Our approach, Uni²Det, is compared against two main baselines. First, we benchmark it against the "Source Only" model, which only uses the source domain for training, and against a robust generalization method, SN (Wang et al., 2020), within the single-source generalization setting. Additionally, we extend the comparison to the dual-source generalization scenario, evaluating Uni²Det against simple data-merging strategies and Uni3D, along with its variants. We also attempt to integrate SN into our method with extra statistical supervision on the target domain. As shown in Table 1, our Uni²Det is proved to achieve more generalized representations on a single dataset as well, further enhancing performance based on SN. This is because our universal framework effectively utilizes the prior information associated with target datasets so as to perform better adaptation. For the dual-source generalization, we discover the trend that using a single detection head in Uni3D yields superior generalization performance compared to employing multiple detection heads. This suggests that training the detection head

Table 1: Results of zero-shot evaluation on unseen datasets. We report the results of the Car (Vehicle on Waymo) category under the IoU threshold of 0.7 and utilize $AP_{BEV}/AP_{3D}$ over 40 recall positions on KITTI. Source Only denotes that the model is trained on the source domain and directly tested on the target domain. S.H. for Uni3D (Zhang et al., 2023) indicates using a single head instead of dataset-specific heads as a variant.

| Single-Source Generalization | | | Dual-Source Generalization | | |
|---|---|---|---|---|---|
| Method | Waymo → KITTI | | Method | Waymo + nuScenes → KITTI | |
| | Detector | mAP | | Detector | mAP |
| Source Only | PV-RCNN | 61.18 / 22.01 | Data Merging | PV-RCNN | 69.07 / 36.17 |
| SN | PV-RCNN | 69.92 / 60.17 | Data Merging (w/ SN) | PV-RCNN | 72.43 / 59.91 |
| **Uni²Det (w/ SN)** | PV-RCNN | **72.41 / 63.96** | Uni3D | PV-RCNN | 71.46 / 37.83 |
| Source Only | Voxel-RCNN | 64.88 / 19.90 | Uni3D (w/ M.H) | PV-RCNN | 71.79 / 38.82 |
| SN | Voxel-RCNN | 75.83 / 55.50 | Uni3D (w/ S.H., SN) | PV-RCNN | 73.48 / 60.51 |
| **Uni²Det (w/ SN)** | Voxel-RCNN | **76.34 / 57.85** | **Uni²Det** | PV-RCNN | 72.39 / 40.12 |
| Method | nuScenes → KITTI | | **Uni²Det (w/ SN)** | PV-RCNN | **75.57 / 64.09** |
| | Detector | mAP | Data Merging | Voxel-RCNN | 69.02 / 36.57 |
| Source Only | PV-RCNN | **68.15** / 37.17 | Data Merging (w/ SN) | Voxel-RCNN | 72.32 / 52.94 |
| SN | PV-RCNN | 60.48 / 49.47 | Uni3D | Voxel-RCNN | 72.68 / 39.65 |
| **Uni²Det (w/ SN)** | PV-RCNN | 66.75 / **55.43** | Uni3D (w/ M.H) | Voxel-RCNN | 73.12 / 40.57 |
| Source Only | Voxel-RCNN | 69.41 / 33.48 | Uni3D (w/ S.H., SN) | Voxel-RCNN | 75.69 / 53.46 |
| SN | Voxel-RCNN | 67.05 / 48.06 | **Uni²Det** | Voxel-RCNN | 74.07 / 43.76 |
| **Uni²Det (w/ SN)** | Voxel-RCNN | **71.02 / 50.94** | **Uni²Det (w/ SN)** | Voxel-RCNN | **78.63 / 58.24** |

across multiple domains enhances its generalization ability and mitigates overfitting to any single domain. Building on this insight, our Uni²Det framework employs multi-stage prompts to enable more unified and universal training, which not only boosts zero-shot generalization performance but also maintains a competitive edge in in-domain tasks. By comparing the overall results, we confirm that Uni²Det significantly improves generalization performance when incorporating new datasets, demonstrating its potential for robust 3D detection across varying domains.

## 4.3 RESULTS OF MULTI-DATASET 3D OBJECT DETECTION

To evaluate the unified design of our framework, we conduct experiments on the two-dataset combination of three widely-used autonomous driving datasets: Waymo (Sun et al., 2020), KITTI (Geiger et al., 2012), and nuScenes (Caesar et al., 2020), and report our results in Table 2 with comparison to baselines demonstrated in (Zhang et al., 2023). We summarize our findings in three points.

Firstly, performance improvement from multi-dataset training is guaranteed for Uni²Det. In some cases, the previous state-of-the-art Uni3D shows worse results under multi-dataset training than when trained on a single dataset (*e.g.*, results on Waymo under Waymo-nuScenes consolidation), indicating that Uni3D did not fully and effectively utilize data from multiple datasets for training. In contrast, our Uni²Det avoids this issue and ensures improved performance with additional datasets for training, demonstrating excellent scalability.

Secondly, a dataset-agnostic head is feasible instead of the dataset-specific head. Although the improved results of Uni3D compared to simply merging datasets demonstrate the benefits of learning dataset-specific detection heads, our work addresses the poor performance issue with a single detection head through a unified paradigm, proving the feasibility of learning a dataset-agnostic detection head. Using more training data from different datasets to train a single detection head in our unified paradigm is more likely to enhance detection performance.

Lastly, Uni²Det is considered a more robust and unified framework for multi-dataset training. Across all dataset combinations, our Uni²Det consistently outperforms Uni3D, demonstrating the effectiveness of our approach in a multi-dataset setting. This result also indicates that our unified training paradigm is stable and robust, capable of converting point cloud data from any source domains or datasets into a more unified distribution through prompts for better prediction. However, our method shows lower AP when inferring some categories (*e.g.*, results for Car on nuScenes under KITTI-nuScenes consolidation). Despite this, considering the boost from other categories within the dataset, it maintains an overall improvement across each dataset.

Table 2: Results of joint training on different dataset consolidations. Following Uni3D (Zhang et al., 2023), we report the car (Vehicle on Waymo), pedestrian, and cyclist results under the IoU threshold of 0.7, 0.5, and 0.5, respectively, and utilize AP/APH of LEVEL 1 metric on Waymo, and $AP_{BEV}/AP_{3D}$ over 40 recall positions on nuScenes and KITTI. P.T. indicates pre-training the baseline detector on the other dataset, and fine-tune the detector on the current dataset. D.M. stands for simply merging data from different datasets as the training data. The best detection results are marked using **bold**. Here we adopt Voxel-RCNN as the baseline detector. Due to the page limitation, we present the experimental results based on PV-RCNN in Appendix.

| | | **Waymo-nuScenes Consolidation** | | | | | | | |
|---|---|---|---|---|---|---|---|---|---|
| Trained on | Method | Tested on Waymo | | | | Tested on nuScenes | | | |
| | | Vehicle | Pedestrian | Cyclist | mAP | Car | Pedestrian | Cyclist | mAP |
| Waymo | w/ P.T. | 75.46/74.99 | 74.58/68.06 | 65.92/64.98 | 71.99/69.34 | 34.34/21.95 | 2.84/1.57 | 0.09/0.02 | 12.42/7.85 |
| nuScenes | w/ P.T. | 6.11/5.90 | 0.77/0.56 | 0.01/0.01 | 2.30/2.16 | 55.23/39.14 | 23.65/16.47 | 8.51/5.80 | 29.13/20.47 |
| Both W&N | D.M. | 66.67/66.23 | 60.36/54.08 | 52.03/51.25 | 59.69/57.19 | 51.40/31.68 | 15.04/9.99 | 5.40/3.87 | 23.95/15.18 |
| | Uni3D | 75.26/74.77 | 75.46/68.75 | 65.02/63.12 | 71.91/68.88 | 60.18/**42.23** | 30.08/24.37 | 14.60/12.32 | 34.95/26.31 |
| | **Uni²Det** | **76.13/75.66** | **77.27/71.84** | **66.40/65.46** | **73.27/70.99** | **60.26**/41.84 | **31.17/25.31** | **17.17/14.42** | **36.20/27.19** |
| | | **KITTI-nuScenes Consolidation** | | | | | | | |
| Trained on | Method | Tested on KITTI | | | | Tested on nuScenes | | | |
| | | Car | Pedestrian | Cyclist | mAP | Car | Pedestrian | Cyclist | mAP |
| KITTI | w/ P.T. | 89.90/81.25 | 59.49/56.17 | 54.55/54.15 | 67.98/63.86 | 12.89/5.52 | 0.24/0.18 | 0.05/0.03 | 4.39/1.91 |
| nuScenes | w/ P.T. | 71.61/40.64 | 39.67/29.99 | 7.29/6.88 | 39.52/25.84 | 53.57/39.65 | 24.93/21.17 | 11.42/9.95 | 29.97/23.59 |
| Both K&N | D.M. | 89.24/73.72 | 61.03/54.55 | 62.71/59.92 | 70.99/62.73 | 41.88/20.48 | 12.58/8.32 | 1.77/0.97 | 18.74/9.92 |
| | Uni3D | 90.09/83.10 | 62.99/58.30 | **70.20/68.10** | 74.43/69.83 | **59.25/41.51** | 29.12/23.18 | 15.16/13.16 | 34.51/25.95 |
| | **Uni²Det** | **90.60/84.16** | **68.40/64.47** | 68.74/65.68 | **75.91/71.44** | 58.09/39.68 | **31.10/25.83** | **20.56/17.53** | **36.58/27.68** |
| | | **KITTI-Waymo Consolidation** | | | | | | | |
| Trained on | Method | Tested on KITTI | | | | Tested on Waymo | | | |
| | | Car | Pedestrian | Cyclist | mAP | Vehicle | Pedestrian | Cyclist | mAP |
| KITTI | w/ P.T. | 89.51/81.41 | 60.30/57.10 | 55.53/51.34 | 68.45/63.28 | 8.70/8.62 | 19.14/16.01 | 21.87/20.83 | 16.57/15.15 |
| Waymo | w/ P.T. | 64.84/19.99 | 62.58/59.01 | 56.44/49.43 | 61.29/42.81 | 72.76/72.26 | 72.42/64.94 | 63.27/62.23 | 69.48/66.48 |
| Both K&W | D.M. | 74.53/32.11 | 60.11/54.85 | 59.69/55.94 | 64.78/47.63 | 74.35/73.85 | 74.80/68.39 | 64.87/63.95 | 71.34/68.73 |
| | Uni3D | 90.03/82.39 | 62.51/57.01 | 69.52/66.30 | 74.02/68.57 | 74.83/74.33 | 74.79/68.24 | 66.83/**65.82** | 72.15/69.46 |
| | **Uni²Det** | **90.30/84.23** | **64.30/61.03** | **71.15/69.18** | **75.25/71.48** | **75.35/74.77** | **76.64/71.22** | **67.03**/65.73 | **73.01/70.57** |

Table 3: Results of jointly training the Voxel-RCNN on three datasets.

Table 4: Ablation study of prompts in different stages of Uni²Det based on Voxel-RCNN.

| Trained on | Tested on | | | |
|---|---|---|---|---|
| | KITTI | NuScenes | Waymo | Avg. |
| KITTI | 70.04/66.09 | 3.84/1.58 | 12.69/11.47 | 28.86/26.38 |
| nuScenes | 32.64/17.70 | 28.99/22.20 | 14.63/14.04 | 25.42/17.98 |
| Waymo | 64.00/45.27 | 12.38/6.34 | 71.84/69.23 | 49.41/40.28 |
| Uni3D | 72.19/67.46 | **35.06/26.48** | 71.95/69.28 | 59.73/54.41 |
| Uni²Det | **76.04/72.61** | 34.03/25.44 | **72.45/70.20** | **60.84/56.08** |

| Method | Voxelization | Backbone | Head | KITTI | Waymo |
|---|---|---|---|---|---|
| Baseline | | | | 72.73/69.94 | 71.09/69.12 |
| Ours | ✓ | | | 73.84/70.56 | 71.71/69.73 |
| | | ✓ | | 73.43/70.19 | 71.52/69.43 |
| | | | ✓ | 73.95/70.43 | 71.73/69.82 |
| | ✓ | ✓ | | 74.96/70.95 | 72.29/69.93 |
| | ✓ | ✓ | ✓ | **75.25/71.48** | **73.01/70.57** |

## 4.4 FURTHER ANALYSIS

**Results on Waymo-KITTI-nuScenes Consolidations.** Table 3 shows the results of jointly training Voxel-RCNN on Waymo-KITTI-nuScenes consolidations. We report average AP over all categories within the dataset. Our Uni²Det demonstrates high detection results on KITTI and Waymo, on which the results of Uni3D do not differ much from those on the single dataset. Overall, more balanced and consistent boosts on different datasets can be observed in Uni²Det on average $AP_{BEV}$ and $AP_{3D}$.

**Ablation on prompts in different stages.** We investigate the influence of prompts in different stages of Uni²Det, including voxelization, backbone, and head. The evaluation setting follows the KITTI-Waymo consolidation in Table 2. As shown in Table 4, the prompt modules implemented at each stage enhance performance compared to the baseline. Notably, the point distribution correction at the voxelization stage shows the most significant improvement among all stages. This demonstrates that learning unified low-level point features is crucial for MDT and serves as a foundation for subsequent prompting stages. We also observe that gradually adding prompts stage-by-stage results in noticeable performance gains for each stage, reflecting the complementarity between the prompting modules. At last, using prompts from all stages together achieves a significant improvement compared to the baseline.

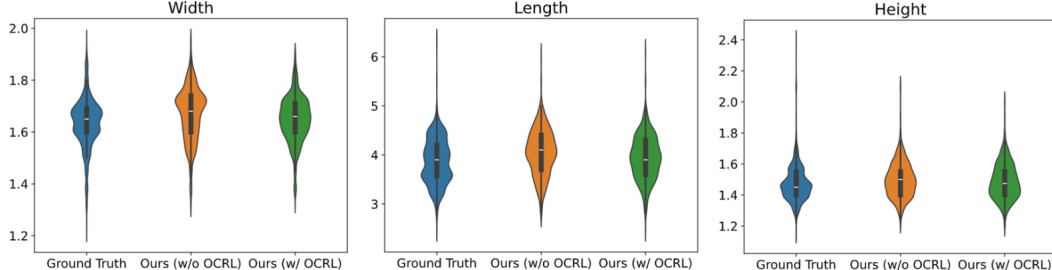

Figure 4: Illustration of statistical distribution differences of object size (length, width, and height) in KITTI between the ground truth and the predictions of Uni$^2$Det with and without object-conditional residual learning (OCRL) module.

**Ablation on object-conditional residual learning.** We investigate the statistical distribution differences of object size in KITTI between the ground truth and the predictions of Uni$^2$Det with and without object-conditional residual learning (OCRL) module. As shown in Figure 4, it is evident that our method, particularly with the OCRL module, more closely approximates the ground truth distribution, especially in terms of mean and variance, thereby reducing statistical differences across datasets and mitigating potential performance degradation. However, this module may also misidentify the discriminative domain for samples from out-of-domain scenes in certain cases, leading to ambiguous residuals that may cause inaccurate bounding box predictions.

**Ablation on normalization strategies.** We conduct ablation studies of normalization strategies in PPT (Wu et al., 2024), Uni3D (Zhang et al., 2023) and our Uni$^2$Det on the Waymo-KITTI consolidation based on Voxel-RCNN, as shown in Table 5. There are no other well-designed modules and only with a single detection head for fair comparison. Our proposed mean-shifted batch normalization is more advantageous, especially on KITTI where only the front view is provided, and it is important to note that the other two compared methods are not conducive to generalization scenarios, which further highlights our strengths.

Table 5: Ablation on normalization strategies based on Voxel-RCNN.

| Method | KITTI | Waymo |
|---|---|---|
| Baseline | 72.73/69.94 | 71.09/69.12 |
| Uni3D | 73.01/70.12 | 71.45/69.23 |
| PPT | 73.25/70.09 | **71.73**/69.46 |
| Uni$^2$Det | **73.84/70.56** | 71.71/**69.73** |

## 5 CONCLUSION

We introduce Uni$^2$Det, a novel framework designed for unified and universal multi-dataset training in 3D detection which utilizes multi-stage prompting modules to harmonize differences between datasets by leveraging dataset-specific characteristics, ensuring robust performance across various domains and effective generalization to new domains. Our work is promising to enhance performance across various domains and facilitate effective generalization to new ones in 3D detection, which has the potential to advance fields like autonomous driving.

**Limitation and discussion.** We identify a limitation in our approach, which relies on the assumption of an identical set of categories across different datasets. This limitation impedes the detection of diverse label spaces with more varied categories. We propose that refining categories by subdividing objects based on size to create a unified label space or mapping unseen classes to existing categories using prior knowledge (e.g., object size) both facilitate zero-shot generalization. Additionally, open-vocabulary 3D detection, enhanced by textual prompts, can expand the range of detectable categories. We expect that future research building on our work will contribute to the development of unified, universal 3D detection systems. Moreover, our method has not yet been applied to indoor datasets, and we aim to make it more versatile and effective across diverse applications.

## 6 ACKNOWLEDGMENT

This work was supported by National Natural Science Fund of China (No.62473286).

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

# APPENDIX

## A    EXTRA EXPERIMENTS

**Results of multi-dataset 3D object detection based on PV-RCNN.**    We present the experimental results for the two-dataset combination of three autonomous driving datasets using PV-RCNN, as shown in Table 6. The conclusions are consistent with those in Table 2, further demonstrating that Uni$^2$Det is a more robust and unified framework for multi-dataset training.

Table 6: Results of joint training on different dataset consolidations based on PV-RCNN. The experiment and evaluation settings follow Table 2.

| | | **Waymo-nuScenes Consolidation** | | | | | | | |
|---|---|---|---|---|---|---|---|---|---|
| Trained on | Method | Tested on Waymo | | | | Tested on nuScenes | | | |
| | | Vehicle | Pedestrian | Cyclist | mAP | Car | Pedestrian | Cyclist | mAP |
| Waymo | w/ P.T. | 74.77/74.26 | 73.32/66.31 | 64.06/63.05 | 70.72/67.87 | 33.86/17.47 | 2.88/1.53 | 0.04/0.01 | 12.26/6.34 |
| nuScenes | w/ P.T. | 44.59/44.24 | 7.67/6.33 | 8.77/8.58 | 20.34/19.72 | 57.92/41.53 | 24.32/17.31 | 11.52/9.19 | 31.25/22.68 |
| Both W&N | D.M. | 66.22/65.75 | 55.41/49.29 | 56.50/55.48 | 59.38/56.84 | 48.67/30.43 | 12.66/8.12 | 1.67/1.04 | 21.00/13.20 |
| | Uni3D | 75.54/74.90 | 74.12/66.90 | 63.28/62.12 | 70.98/67.97 | 60.77/42.66 | 27.44/21.85 | 13.50/11.87 | 33.90/25.46 |
| | **Uni$^2$Det** | **76.03/75.53** | **76.24/70.29** | **64.97/63.95** | **72.41/69.92** | **61.38/42.76** | **28.60/22.49** | **15.10/12.90** | **35.03/26.05** |
| | | **KITTI-nuScenes Consolidation** | | | | | | | |
| Trained on | Method | Tested on KITTI | | | | Tested on nuScenes | | | |
| | | Car | Pedestrian | Cyclist | mAP | Car | Pedestrian | Cyclist | mAP |
| KITTI | w/ P.T. | 89.26/83.14 | 60.56/55.90 | 63.60/62.88 | 71.14/67.31 | 13.43/5.61 | 0.69/0.27 | 0.04/0.00 | 4.72/1.96 |
| nuScenes | w/ P.T. | 69.40/38.25 | 33.24/24.88 | 1.68/1.61 | 34.77/21.58 | 53.24/36.72 | 20.65/17.09 | 8.95/7.58 | 27.61/20.46 |
| Both K&N | D.M. | 87.79/77.95 | 55.52/48.29 | 59.15/55.10 | 67.49/60.45 | 41.29/21.57 | 10.21/7.08 | 1.23/1.15 | 17.58/9.93 |
| | Uni3D | 89.77/85.49 | 60.03/55.58 | 69.03/66.10 | 72.94/69.06 | **59.08/41.67** | 25.27/19.26 | 12.26/10.83 | 32.20/23.92 |
| | **Uni$^2$Det** | **90.52/85.36** | **61.73/58.53** | **71.76/69.29** | **74.67/71.06** | 58.30/41.21 | **29.11/24.00** | 12.62/10.93 | **33.34/25.38** |
| | | **KITTI-Waymo Consolidation** | | | | | | | |
| Trained on | Method | Tested on KITTI | | | | Tested on Waymo | | | |
| | | Car | Pedestrian | Cyclist | mAP | Vehicle | Pedestrian | Cyclist | mAP |
| KITTI | w/ P.T. | 89.40/83.42 | 62.69/58.86 | 59.96/59.43 | 70.68/67.24 | 8.75/8.64 | 12.12/9.90 | 9.20/8.76 | 10.02/6.10 |
| Waymo | w/ P.T. | 69.25/25.91 | 59.16/55.92 | 56.09/50.50 | 61.50/44.11 | 71.08/70.54 | 70.12/62.91 | 62.37/61.40 | 67.86/64.95 |
| Both K&W | D.M. | 87.49/68.35 | **62.84/60.06** | 68.09/65.75 | 72.81/64.72 | 50.68/50.31 | 58.76/52.59 | 55.14/54.17 | 54.86/52.36 |
| | Uni3D | 89.42/83.15 | 60.85/57.49 | 71.61/65.88 | 73.96/68.84 | 75.07/74.54 | 72.95/66.08 | 63.80/62.92 | 70.61/67.85 |
| | **Uni$^2$Det** | **90.70/84.65** | 61.02/58.33 | **72.86/71.26** | **74.86/71.41** | **75.43/74.92** | **74.96/69.20** | **65.57/64.49** | **71.99/69.54** |

**Ablation on the balancing ratio.**    We conduct ablation studies regarding the hyperparameter balancing ratio $\alpha$ based on different 3D detectors, based on AP$_{3D}$ metrics of KITTI on the Waymo-KITTI consolidation, as shown in Figure 5. When $\alpha = 0.1$ for PV-RCNN and $\alpha = 0.5$ for Voxel-RCNN, the performance can be optimal. Since Voxel-RCNN mainly relies on voxelized point cloud data and is more sensitive to the distribution of point clouds, the balancing ratio for point feature correction needs to be larger.

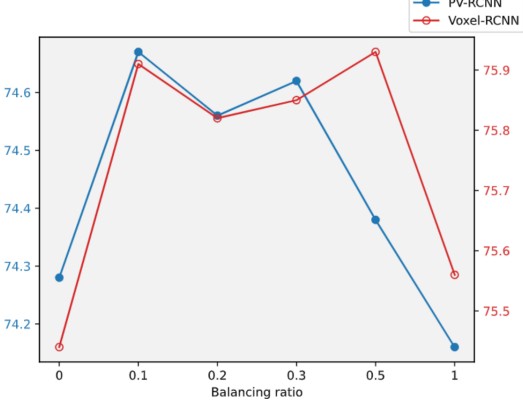

Figure 5: Ablation on the balancing ratio based on PV-RCNN and Voxel-RCNN, based on AP$_{3D}$ metrics of KITTI on the Waymo-KITTI consolidation.

