# OpenReview forum: "Uni$^2$Det: Unified and Universal Framework for Prompt-Guided Multi-dataset 3D Detection"
_ICLR.cc/2025/Conference — ICLR 2025 Poster_

### Official Review · Reviewer_dmVZ · 2024-10-28

**Soundness:** 2
**Presentation:** 2
**Contribution:** 2
**Rating:** 6
**Confidence:** 4

**Summary:**

In this work, the authors propose Uni$^2$Det, a prompt-guided multi-dataset 3D object detection framework. In detail, the authors handle multi-dataset 3DOD from three perspectives (*i.e.*, voxelization, bev features, and detection head). First, the designed point distribution correctness uses instance- and batch-level normalization to deal with the distribution shift in different datasets. Further, the bev-based range masking can select features of corresponding areas of the different datasets. Finally, the authors propose object-conditioned residual learning to inject dataset-specific features into each instance to further decode the bounding boxes. Experiments show that the proposed method can achieve higher results on both in-domain and out-of-domain tasks.

**Strengths:**

- Experiments show that the proposed Uni$^2$Det can achieve SOTA performance on both cross-domain and multi-dataset training settings.
- The authors treat dataset-specific attributes as prompts which is quite interesting.

**Weaknesses:**

1. My first question is about the proposed BEV-based range masking. I wonder about the masking strategy during the inference phase. Do we need to know the dataset-specific point cloud range before evaluation? If so, what will happen when we do not know the point cloud range? Besides, in the domain generalization experiments, if we use the point cloud range of the target domain, it seems unfair since the domain-specific knowledge has already been injected into the model.
2. Further, I wonder about the performance using more recent 3D detection models. I fully understand that the authors chose PV-RCNN and Voxel R-CNN for a fair comparison. However, they have been proposed for several years.
3. Questions about experimental settings, especially hyperparameters. I need to know how the authors define the value of hyperparameters such as $\alpha$, there seem no ablation studies about this.
4. The motivation of point mean-shift normalization and object-conditional residual learning is similar to the data-level correction and semantic-level coupling and recoupling. Both of them first handle data-level shifts and then strengthen domain-specific features. The authors need to explain the advantages of the proposed methods.

**Questions:**

- I know MDF may represent multi-dataset fusion. However, the authors use MDT in the preliminary and then use MDF in the following section. This needs to be modified to avoid confusion.

---

> ### Author Response · Authors · 2024-11-21
>
> Thank you for your patient and detailed review. We try to address your comments below.
> ### ***W1: Discuss details of the BEV-based range masking.***
>
> It’s essential to clarify that the dataset-specific point cloud range is **a critical prerequisite** for accurately applying the masking strategy and prompting the detector. In fact, the goal of our work is to inject intrinsic attributes of datasets (such as the point cloud range), which do not require learning, into the model. It is generally assumed that **these attributes are also known when applying to new datasets**, thereby enabling the model to generalize better. In cases where the point cloud range is unknown, **it’s feasible to use statistical methods to estimate it** based on point cloud distribution characteristics of the target domain, thus providing an approximate range for masking.
>
> ### ***W2: Experiments using more recent 3D detection models are needed.***
>
> As a plug-and-play method, **each module in our method can be** **easily transferred to other 3D detectors and can be effective.** We apply our method to a more recent 3D detector, PV-RCNN++, which is a more advanced 3D detector with better performance compared with detectors in our paper. We conduct experiments on the Waymo-nuScenes consolidation. The evaluation metrics are identical to our paper. The results validate that our method still maintains a robust performance improvement on more advanced 3D detectors.
>
> | Method       | W-Vehicle   | W-Pedestrian | W-Cyclist   | W-mAP       | N-Car       | N-Pedestrian | N-Cyclist   | N-mAP       |
> | :----------- | :---------- | :----------- | :---------- | :---------- | :---------- | :----------- | :---------- | :---------- |
> | Data Merging | 70.71/69.88 | 65.31/60.76  | 57.73/56.94 | 64.58/62.53 | 52.13/31.46 | 19.02/15.67  | 4.68/3.91   | 25.28/17.01 |
> | Uni3D        | 77.08/76.62 | 78.15/71.29  | 66.88/64.67 | 74.04/70.86 | 62.13/42.57 | 32.94/26.53  | 22.03/19.74 | 39.03/29.61 |
> | Uni2Det      | 78.26/77.98 | 78.87/73.12  | 68.46/66.53 | 75.20/72.54 | 62.75/42.76 | 34.45/27.94  | 25.26/21.30 | 40.82/30.67 |
>
> (Here W- denotes Waymo and N- denotes nuScenes.)
>
> ### ***W3: Discuss the value of hyperparameters like $\alpha$.***
>
> We have already provided an ablation study on the parameter of balancing ratio α in the appendix of the paper, but we notice that we mistakenly wrote $\alpha$ as $\lambda$ there, for which we deeply apologize. **Essentially, $\alpha$ is a hyperparameter related to the backbone.**  As for the two backbones we are using, since Voxel-RCNN mainly relies on voxelized point cloud data and is more sensitive to the distribution of point clouds, the balancing ratio for point feature correction needs to be larger.
>
> ### ***W4: Explain the advantages of the proposed methods, including point mean-shift normalization and object-conditional residual learning.***
>
> **Point mean-shift normalization** addresses data differences in point cloud distribution. Unlike standard batch normalization, which assumes that all frames or instances share a similar distribution, **our frame-level mean shifts allow for instance-specific statistics, recognizing that data from different frames may exhibit distinct characteristics due to sensor discrepancies or other variations, which is highly correlated with dataset’s properties.** This approach helps regularize samples within each dataset, enabling the capture of unique dataset attributes while maintaining normalized features across datasets. By capturing unique characteristics of each frame, frame-level mean shifts reduce reliance on a single dataset-level distribution, as cumulative frame adjustments capture broader dataset variability.
>
> **Object-conditional residual learning** addresses object size distribution differences by applying feature adjustments to each object's characteristics, enhancing adaptability across diverse datasets. Unlike semantic-level coupling and recoupling, which is not well-generalized in out-of-domain settings, **our method reduces distribution gaps by adjusting features specifically for each object, improving detection accuracy and transferability to new domains with learnable residuals.** By incorporating object-specific prompts, it enables a universal detection head that effectively handles varied datasets, achieving a more unified and flexible detection framework.
>
> ### ***W5: Explanation of referencing errors of MDT.***
>
> You are correct that there was inconsistency in the terminology and the acronym 'MDF' was mistakenly used. **In fact, the correct term should all be 'MDT', which stands for Multi-Dataset Training.** We will revise the paper to ensure consistency by using 'MDT' throughout.

---

> > ### Comment · Reviewer_dmVZ · 2024-11-23
> > **Decision after rebuttal**
> >
> > I appreciate the authors' effort in the rebuttal phase and they have addressed most of my concerns. I will slightly improve my score to 6.

---

### Official Review · Reviewer_C7SQ · 2024-10-29

**Soundness:** 3
**Presentation:** 2
**Contribution:** 3
**Rating:** 6
**Confidence:** 4

**Summary:**

This paper proposes a multi-stage prompting pipeline to use multiple data sources to train one 3D detector. Specifically, they performed mean-shifted batch normalization, applied BEV-based masking, and trained a discriminator on object-conditioned residual using classification loss. The object-conditioned residual is added to the RoI feature for prediction.

**Strengths:**

1. The paper is easy to understand and has a clear structure.
2. De-meaning data across batches and within one instance is an interesting way to improve generalization. Using dataset-relevant information as prompts is also very interesting.
3. The performance seems good.

**Weaknesses:**

There are too few classes for evaluation, and the class domain is limited to autonomous driving scenarios. The authors mentioned this in section 3.1.

**Questions:**

1. In mean-shifted batch normalization, it would be helpful to further explain how to calculate the instance-specific mean. Additionally, it will be wonderful to compare the proposed method with an alternate approach of first de-mean data using instance-specific mean, following batch-wise demeaning without alpha weighting. Further more, what is the influence of completely discarding this stage, only using the backbone and the head?
2. In object-conditional residual learning, how is the dataset id encoded in training? Also, during testing on zero-shot datasets, does this pipeline obtain object-conditional residual before prediction?
3. In testing, it will be interesting to see how the model performs on classes that do not intersect with existing class vocabulary. At the same time, applying this method to datasets with various scenes, i.e., indoor 3D detection datasets, could provide a broader perspective on its robustness.

---

> ### Author Response · Authors · 2024-11-21
>
> Thank you for your patient and detailed review. We try to address your comments below.
> ### ***W1: Discuss how to calculate the instance-specific mean and compare the proposed method with different normalization settings.***
>
> For the instance-specific mean, **we directly average features of points within one frame (instance).** By capturing unique characteristics of each frame, frame-level mean shifts reduce reliance on a single dataset-level distribution, as cumulative frame adjustments capture broader dataset variability. **The approaches of first de-meaning data using instance-specific mean and batch-wise de-meaning without alpha weighting** **correspond to $\alpha=1$ and $\alpha=0$, which we have conducted experiments in the appendix** (We notice that we mistakenly wrote α as λ there, for which we deeply apologize). In fact, to completely discard this stage of prompt is just using batch-wise de-meaning without alpha weighting, which is included in the baseline models.
>
> ### ***W2: Discuss details of the object-conditional residual learning.***
>
> **We use dataset id as the groundtruth label for training the discriminator as common classification.** During testing on zero-shot datasets, **we discard the discriminator and keep the residual function to output the object-conditional residual before prediction.** By this approach, when encountering a sample from an unseen domain, we can analyze it and identify this domain that closely resembles one of the source domains, then generate the similar residual, which enhances the model's generalization capability.
>
> ### ***W3: Discuss how the model performs on classes that do not intersect with existing class vocabulary and how to apply this method to datasets with various scenes.***
>
> **Diverse, non-overlapping, or missing categories**: We agree that it's essential to extend Uni$^2$Det to handle datasets on classes that do not intersect with existing class vocabulary. In fact, this is precisely the goal of our ongoing extension work. There are multiple ways we plan to achieve this.
>
> - For datasets with diverse but partially overlapping label spaces, we can **refine the category labels by subdividing objects based on size** (e.g., categorizing "vehicle" as "small," "medium," or "large"), thus creating a unified label space across datasets and supporting generalization to new datasets with a more detailed label space.
>
> - For datasets with non-overlapping labels, we could **map unseen classes to existing ones using prior knowledge** of distributional similarities of pre-defined properties (e.g., object size), enabling zero-shot generalization.
>
> - **Open-vocabulary 3D object detection techniques** also provide a promising approach to further broaden the range of categories we can handle. Incorporating textual information as prompts could greatly enhance the model's open-vocabulary 3D object detection capabilities. By using descriptive text prompts (e.g., "a large vehicle with four wheels"), the model can associate physical characteristics with categories it hasn’t seen before, enabling zero-shot recognition. This approach **leverages semantic information to help the model generalize to new classes**, expanding its detection scope without needing additional labeled data in real-world applications where unfamiliar objects are frequently encountered.
>
> While the current version of our paper follows prior work by assuming consistent categories, **we will discuss these additional strategies in the final version to clarify our approach and its potential for more diverse scenarios.**
>
> **Potential of applying this method to various scenes like indoor datasets:** Uni$^2$Det is primarily designed and optimized for outdoor datasets, especially with components like the BEV-based range masking module, which are tailored for outdoor sensor setups. As a result, performance on indoor datasets such as ScanNet or SUN-RGBD may not fully reflect the strengths of our approach, particularly due to the differences in sensor specifications and data characteristics between indoor and outdoor environments. However, we believe that **other modules in our framework, particularly those that focus on cross-domain generalization, could still provide valuable contributions to object detection in indoor datasets.** Future work may explore adapting our approach to better handle the unique challenges posed by indoor sensor setups, enhancing its robustness across various sensor configurations.

---

> > ### Comment · Reviewer_C7SQ · 2024-11-23
> > **Thank the authors for the response**
> >
> > Dear Authors,
> > I have read your replies carefully, and they have addressed my existing concerns with mean-shifted batch normalization and residual learning. For the limitations, I agree with reviewer sQWH's comment on 'Assumption of Identical Categories'. This keeps my rate at 6. I encourage the authors to provide evaluations on more categories for main results in future work.

---

### Official Review · Reviewer_sQWH · 2024-10-30

**Soundness:** 3
**Presentation:** 3
**Contribution:** 2
**Rating:** 8
**Confidence:** 2

**Summary:**

The paper introduces Uni2Det, a framework designed to tackle the challenges of 3D object detection across multiple datasets. By leveraging multi-stage prompting modules, Uni2Det aims to overcome data distribution discrepancies and enhance generalization capabilities. The framework is evaluated on datasets such as KITTI, Waymo, and nuScenes, demonstrating superior performance compared to existing methods.

**Strengths:**

(1) Innovative Techniques:

The introduction of multi-stage prompting modules is a novel approach to address the differences in data distribution among various datasets, potentially setting a new standard for unified 3D detection frameworks.


(2) Comprehensive Experiments:

The authors conducted extensive experiments across multiple prominent datasets, showing significant improvements in both multi-dataset training scenarios and zero-shot transfer performance. This thorough evaluation strengthens the validity of the proposed method.

(3) Robust Generalization:

The framework shows impressive zero-shot capabilities, effectively adapting to unseen datasets without additional retraining. This highlights its potential for real-world applications where models need to perform well across diverse environments.

(4) Scalability:

Uni2Det demonstrates improved scalability by effectively utilizing additional datasets for training, ensuring performance boosts without the pitfalls of negative transfer that other methods might suffer from.

**Weaknesses:**

(1) Complexity and Implementation:

The multi-stage prompting modules, while innovative, may introduce additional complexity to the framework. This could potentially affect the ease of implementation and deployment in practical scenarios, requiring more resources and expertise.

(2) Assumption of Identical Categories:

The framework relies on the assumption that datasets share identical categories, which limits its applicability to datasets with diverse or non-overlapping label spaces. This constraint could hinder the adoption of Uni2Det in broader applications.

(3) Limited Component Analysis:

The paper could benefit from a more detailed analysis of how each component of the framework contributes to overall performance. Understanding the impact of each module could provide deeper insights and guide future improvements.

(4) Computational Overhead:

The potential increase in computational requirements due to the additional modules is not thoroughly discussed. This could be a concern for deploying the framework in resource-constrained environments.

**Questions:**

(1) Sensor Variability:

How does Uni2Det perform when dealing with datasets that utilize sensors with significantly different specifications? Understanding this could provide insights into its robustness across various sensor setups. For example, how is the performance on indoor datasets like Scannet [2] or SUN-RGBD [1].

(2) Extension to Diverse Categories:

Is it possible to extend the framework to handle datasets with richer categories, like some open-vocabulary 3d object detection works [3, 4]? This would enhance its applicability to a wider range of scenarios. Please provide discussions in the paper.

(3) Efficiency Considerations:

What are the computational overhead and latency introduced by the multi-stage prompting modules? Providing metrics on efficiency would be beneficial for assessing practical deployment scenarios.

(4) Failure Case Analysis:

Can the authors provide examples or analysis of scenarios where the framework might underperform? This would help in understanding its limitations and potential areas for improvement. Please include failure cases in the paper.

[1] Shuran Song, Samuel P Lichtenberg, and Jianxiong Xiao. Sun rgb-d: A rgb-d scene understanding benchmark suite. In CVPR, 2015.

[2] Angela Dai, Angel X Chang, Manolis Savva, Maciej Halber, Thomas Funkhouser, and Matthias Nießner. Scannet: Richly-annotated 3d reconstructions of indoor scenes. In CVPR, 2017.

[3] Y. Lu, C. Xu, X. Wei, X. Xie, M. Tomizuka, K. Keutzer, and S. Zhang, “Open-vocabulary point-cloud object detection without 3d annotation,” CVPR, 2023.

[4] Y. Cao, Y. Zeng, H. Xu, and D. Xu, “Coda: Collaborative novel box discovery and cross-modal alignment for open-vocabulary 3d object detection,” in NeurIPS, 2023.

---

> ### Author Response · Authors · 2024-11-21
>
> Thank you for your patient and detailed review. We try to address your comments below.
> ### ***W1: Discuss complexity and implementation in practical scenarios, as well as the potential increase in computational requirements when facing practical deployment scenarios.***
>
> We acknowledge the concern about potential complexity and computational overhead introduced by our multi-stage prompting modules. Our design, however, intentionally emphasizes **simplicity and efficiency** to keep the framework practical for real-world deployment.
>
> - **Prompt for Voxelization:** In this module, we calculate mean point cloud features and integrate them directly into Batch Normalization layers. This integration is computationally efficient, as it leverages standard operations within the Batch Normalization without adding any substantial processing overhead.
>
> - **Prompt for Backbone:** The BEV-based range masking method we employ relies only on the point cloud range to create a 2D mask. This masking process is lightweight, as it does not involve any complex transformations or additional modules, thus ensuring that computational demand remains minimal.
>
> - **Prompt for Head:** For this module, we use a simple residual function and discriminator, implemented via linear layers. By designing the prompt module this way, we avoid the need for multiple detection heads, significantly reducing the parameter count. This approach minimizes the computational burden on the final detection layers, keeping the module streamlined and resource-efficient.
>
> We also demonstrate running time comparison for the baseline and our method averaged on the test set of KITTI based on PV-RCNN. Compared to the baseline method, **our approach introduces almost no computational overhead.**
>
> | Baseline (ms) | Uni$^2$Det (ms) | Overhead Ratio |
> | - | - | - |
> | 97.2          | 97.7            | 0.51%          |
>
> **Overall, these design choices ensure that our framework remains lightweight and manageable, with minimal impact on computational resources and latency, as we retain as many shared parameters across datasets as possible in our framework.** We believe this approach makes our framework suitable for deployment in resource-constrained environments, without requiring extensive expertise or complex implementation efforts for new datasets.
>
> ### ***W2: This approach limits its applicability to datasets with diverse or non-overlapping label spaces. Discuss the potential to extend the framework to handle datasets with richer categories.***
>
> We agree that extending Uni$^2$Det to handle datasets with diverse or non-overlapping label spaces would indeed enhance its applicability. In fact, this is precisely the goal of our ongoing extension work. There are multiple ways we plan to achieve this. Due to the word limit here, please refer to our response to W3 of Reviewer C7SQ, which provides a detailed discussion.
>
> ### ***W3: This paper provides limited component analysis.***
>
> We have conducted ablation experiments in Table 4 of the paper to verify how each component of the framework contributes to overall performance. Additionally, we can provide experimental results for other combinations, as shown in the table below, which provide enough insights into the effectiveness of each component.
>
> |  Method  | Voxelization | Backbone | Head |    KITTI    |    Waymo    |
> | :-: | :-: | :-: | :-: | :-: | :-: |
> | Baseline |              |          |      | 72.73/69.94 | 71.09/69.12 |
> |          |      ✔️       |          |      | 73.84/70.56 | 71.71/69.73 |
> |          |              |    ✔️     |      | 73.43/70.19 | 71.52/69.43 |
> |          |              |          |  ✔️   | 73.95/70.43 | 71.73/69.82 |
> |          |      ✔️       |    ✔️     |      | 74.96/70.95 | 72.29/69.93 |
> |   Ours   |      ✔️       |    ✔️     |  ✔️   | 75.25/71.48 | 73.01/70.57 |
>
> ### ***W4: Provide insights into its robustness across various sensor setups like indoor datasets.***
>
> This is a valuable question. Please refer to our response to W3 of Reviewer C7SQ due to the word limit here.
>
> ### ***W5: Provide examples or analysis of scenarios where the framework might underperform.***
>
> Based on our extensive studies on the multi-dataset training, our Uni$^2$Det’s performance can be affected **in some extreme out-of-domain scenarios** where the dataset attributes, such as point cloud distribution or LiDAR sensor properties, deviate significantly from those in the training datasets. For example, datasets with highly skewed distributions in object size or sensor orientation, not observed during training, could result in performance drops despite our multi-stage prompting modules. **In some cases, the object-level residual learning module cannot identify a discriminative domain for samples from out-of-domain scenes, generating vague residuals that result in inaccurate bounding box predictions.** To address this issue, we will include more detailed failure cases and empirical analysis in the revised manuscript.

---

> > ### Comment · Reviewer_sQWH · 2024-11-25
> > **Further comments**
> >
> > Thank the authors for the reply. Could you please compare this work with the latest multi-domain work [1] in terms of the method and performance?
> >
> > [1] Zhenyu Wang, Yali Li, Hengshuang Zhao, and Shengjin Wang. One for All: Multi-Domain Joint Training for
> > Point Cloud Based 3D Object Detection. In NeurIPS, 2024.

---

> > > ### Author Response · Authors · 2024-11-27
> > >
> > > OneDet3D aims to achieve universality by integrating indoor and outdoor datasets within a unified framework while adopting an open-vocabulary strategy for detection. Similar to Uni$^2$Det, this method seeks to mitigate data-level interference caused by variations in the inherent structure of point clouds. However, a limitation of this approach is that its language-guided classification, based on CLIP embeddings, does not sufficiently address label space conflicts across datasets. Rather than using a relatively broader range of categories for training and inference, as in Uni$^2$Det, their experiments for outdoor datasets are restricted to the car category only.
> > >
> > > Unlike OneDet3D, Uni$^2$Det specifically addresses the challenges of multi-dataset 3D object detection in outdoor scenarios. It also demonstrates potential applicability to indoor datasets owing to its prompt-based generalization capabilities. Uni$^2$Det employs a multi-stage prompting framework to directly address data distribution shifts and modality discrepancies across datasets. This framework, along with internal prompting modules, enable Uni$^2$Det to achieve superior performance on both in-domain and out-of-domain tasks, as demonstrated on benchmark datasets such as KITTI, Waymo, and nuScenes.
> > >
> > > We evaluate the cross-domain performance of these two methods on the outdoor Waymo dataset. The Uni$^2$Det models are trained on the KITTI and nuScenes datasets, using PV-RCNN as its backbone. Notably, Uni$^2$Det surpasses OneDet3D, even when the latter is trained jointly on four datasets.
> > >
> > > | Method     | Trained on                          | AP$^{3D}$ | AP$^{BEV}$ |
> > > | ---------- | ----------------------------------- | --------- | ---------- |
> > > | OneDet3D   | KITTI, nuScenes                     | 40.3      | 61.3       |
> > > | OneDet3D   | SUN RGB-D, ScanNet, KITTI, nuScenes | 41.1      | 61.7       |
> > > | Uni$^2$Det | KITTI, nuScenes                     | 42.8      | 63.3       |
> > >
> > > We will cite this work in the formal version of our paper.

---

> > > > ### Comment · Reviewer_sQWH · 2024-11-29
> > > > **Further comments**
> > > >
> > > > Thanks for the feedback from the authors. I respectfully disagree with the statement: "However, a limitation of this approach is that its language-guided classification, based on CLIP embeddings, does not sufficiently address label space conflicts across datasets." In my experience, CLIP embeddings provide powerful open-vocabulary knowledge which enhances performance on novel categories. However, I acknowledge that this broader open-vocabulary ability may slightly impact performance on specific known categories. That said, the improved performance noted above is reasonable. Furthermore, the lack of emphasis on open-vocabulary ability in this paper is not a critical limitation, as it is not the primary focus of the work. Additionally, other concerns have been addressed well. Therefore, I am raising my score to 8. Gook luck to the authors :)
> > > >
> > > > Besides, I was just curious and asked about the comparison with OneDet3D. According to general conference rules, since NeurIPS has not yet taken place, citing it is not mandatory. The authors can decide whether to cite and discuss it based on the relevance of the paper.

---

### Official Review · Reviewer_AxYi · 2024-11-12

**Soundness:** 3
**Presentation:** 4
**Contribution:** 3
**Rating:** 8
**Confidence:** 4

**Summary:**

This paper addresses a critical challenge in autonomous driving research: integrating multiple datasets collected across different locations and using varied sensor types, especially LiDAR-based detection datasets. Due to differences in distribution, such datasets cannot be directly combined for training without significant performance trade-offs. The authors propose a novel, multi-stage prompting approach to effectively leverage the unique characteristics of each dataset and mitigate modality disparities, ultimately enabling a more robust multi-dataset training framework. This multi-stage approach is designed as a plug-and-play module that enhances several advanced 3D detection frameworks by targeting three stages: voxelization, BEV feature processing, and the detection head. Experimental results substantiate that the proposed multi-dataset training technique significantly improves cross-domain performance relative to baseline models, showcasing its effectiveness in harnessing diverse datasets for training.

**Strengths:**

- The paper tackles a practical issue in autonomous driving, where dataset diversity can restrict the ability to scale training. By addressing how to integrate datasets with different characteristics, the research directly benefits the community by opening up avenues for broader dataset utilization, potentially enhancing the generalization of autonomous driving models.
- The proposed approach is modular, designed as a plug-and-play component compatible with various advanced 3D detectors.
- The use of frame-specific mean shifts for batch normalization is particularly novel. I’m curious whether this concept has been explored in existing literature or if it is an original contribution by the authors.
- The paper includes comprehensive experiments to validate its claims, showing that: (1) the method outperforms baselines like Uni3D, (2) it can be integrated with various 3D detectors, (3) each of the three prompting stages contributes to the improved performance, and (4) OCRL improves the alignment of object size distribution with ground truth data.
- The paper is well-organized and easy to follow, with clear figures. However, providing a list of all abbreviations in figure captions would enhance clarity.

**Weaknesses:**

While the paper is methodologically sound and well-supported by experiments, there are a few areas that could benefit from additional clarity and detail:
- For mean-shifted BN:
1) The authors claim that mean-shifted BN introduces dataset-specific characteristics, yet the parameter \alpha appears to capture only frame-specific characteristics rather than broader dataset-level traits. A more intuitive approach might involve defining dataset-specific means and variances to capture the unique properties of each dataset more accurately. I suggest the authors discuss this alternative and clarify how frame-level mean shifts contribute to dataset-level adaptability.
2) It remains unclear how the parameter \alpha is specified within the model. Further explanation of its determination, whether it is learned, fixed, or computed dynamically, would help readers understand its role in adapting the batch normalization to different datasets.
- For the BEV-based range masking component, the authors choose to concatenate the mask as an additional channel rather than directly masking out unwanted regions in the BEV representation. It would be valuable for the authors to provide a rationale for this choice. Specifically, how does the inclusion of the mask as a separate channel improve model performance or facilitate feature extraction compared to direct masking? A discussion on the impact of this approach on model interpretability and cross-domain generalization would also be insightful.
- More discussion about the limitation when applying the approach to the real-world applications will be helpful.

**Questions:**

Check weaknesses

---

> ### Author Response · Authors · 2024-11-21
>
> Thank you for your patient and detailed review. We try to address your comments below.
> ### ***W1: Discuss dataset-specific means and variances and clarify how frame-level mean shifts contribute to dataset-level adaptability. Explain the parameter $\alpha$ specified within the model.***
>
> **Dataset-specific means and variances:** It should be explained that defining dataset-specific means and variances can indeed leverage the characteristics of the dataset for training, which we have considered before. However, considering the feasibility of generalization to new datasets, **direct access to dataset-level distributions (means and variances) is often unavailable.** Consequently, we use frame-specific characteristics as a substitute.
>
> **Benefits of frame-level mean shifts:** Unlike standard batch normalization, which assumes that all frames or instances share a similar distribution, **our frame-level mean shifts allow for instance-specific statistics, recognizing that data from different frames may exhibit distinct characteristics due to sensor discrepancies or other variations, which is highly correlated with dataset’s properties.** This approach helps regularize samples within each dataset, enabling the capture of unique dataset attributes while maintaining normalized features across datasets. By capturing unique characteristics of each frame, frame-level mean shifts reduce reliance on a single dataset-level distribution, as cumulative frame adjustments capture broader dataset variability.
>
> **Explanation of the parameter of balancing ratio $\alpha$:** We have already provided an ablation study on the parameter of balancing ratio $\alpha$ in the appendix of the paper, but we notice that we mistakenly wrote $\alpha$ as $\lambda$ there, for which we deeply apologize. **Essentially, $\alpha$ is a hyperparameter related to the backbone.**
>
> ### ***W2: Discuss rationale for concatenating the mask as an additional channel and the principle and impact of the approach.***
>
> Concatenating the BEV mask as an additional channel, rather than directly masking out unwanted regions, allows the model to **retain full BEV feature representation while leveraging the mask as contextual information.** This approach ensures that no important spatial data is discarded, enabling the network to **dynamically prioritize relevant areas based on the provided mask.** By keeping the mask as a separate channel, the model can effectively generalize across datasets with varying layouts, preserving both domain-specific and cross-domain features. Additionally, this method enhances interpretability, as it allows for insights into how the network uses the mask to inform its feature extraction and decision-making process, improving transparency and adaptability in diverse environments.
>
> We also provide experimental results of concatenating the BEV mask as an additional channel and directly masking out unwanted regions. We don't introduce other modules for fair comparison.
>
> | Method           | W-Vehicle       | W-Pedestrian    | W-Cyclist       | W-mAP           | N-Car           | N-Pedestrian    | N-Cyclist       | N-mAP           |
> | :- | :- | :- | :- | :- | :- | :- | :- | :- |
> | Baseline         | 74.13/73.86     | 75.31/68.91     | 64.68/62.80     | 71.37/68.52     | 59.78/41.36     | **30.93**/25.01 | 15.73/12.42     | 35.48/26.26     |
> | Directly Masking | 74.26/73.75     | 75.57/69.28     | 64.70/63.01     | 71.51/68.68     | 59.89/41.44     | 30.85/**25.17** | 15.99/13.10     | 35.58/26.57     |
> | Concatenating    | **75.44/74.23** | **75.89/70.12** | **65.45/63.76** | **72.26/69.37** | **59.97/41.54** | 30.67/25.06     | **16.47/13.65** | **35.70/26.75** |
>
> ### ***W3: Discuss the limitation when applying the approach.***
>
> **Diverse, non-overlapping, or missing categories**: A notable challenge is the assumption that datasets share identical categories, which may not always be the case in practice. In real-world scenarios, datasets often feature diverse, non-overlapping, or missing categories, making it difficult for the model to adapt seamlessly across different label spaces. This limitation hinders the model's ability to effectively handle the broader variety of labels present in real-world data, especially in domains with highly specialized or evolving categories.
>
> **Generalization in more dynamic or heterogeneous environments:** Furthermore, the dependency on dataset-specific prompts, while effective for reducing domain disparities, may not be sufficient to fully address the complexity and variability encountered in more dynamic or heterogeneous environments. As a result, future research should aim to refine the model’s ability to generalize across more diverse and ever-changing label sets, making the detection process more inclusive and adaptable. We hope that future developments will address these challenges, enhancing the model's practical applicability and robustness across a wider range of real-world scenarios.

---

### Author Response · Authors · 2024-11-21

The reviewers collectively acknowledged the novelty and practical value of our approach, particularly innovations like point mean-shift normalization and object-conditional residual learning. They suggested clarifying key components such as dataset-specific means and variances, the parameter $\alpha$, and the rationale behind specific design choices. Additionally, they recommended extending experiments with recent models, evaluating performance across diverse datasets, and discussing practical deployment challenges. Addressing these points will enhance the clarity, robustness, and applicability of our method.

---

### Meta-Review · Area_Chair_Fdx6 · 2024-12-22

**Metareview:**

This paper presents a novel unified framework for prompt-guided multi-dataset 3D detection. The proposed method shows very significant performance on several challenging benchmarks including KITTI, Waymo, and nuScenes. All the reviewers are positive about the technical contributions made by this submission. AC also agrees with the comments of the reviewers and recommends an acceptance.

**Additional Comments On Reviewer Discussion:**

The reviewers asked for further clarifications and additional ablation studies. The reviewers are satisfied with the authors' rebuttal.

---

### Decision · Program_Chairs · 2025-01-22

Accept (Poster)